

# Attention behaviours but not pain-related behaviours are modified by the presence of a twin in lambs undergoing castration by rubber ring

Andrew Inhyuk Cho[1,2], Caroline Lee[1] and Alison Small[1]

[1] CSIRO Agriculture & Food, Commonwealth Scientific and Industrial Research Organisation, Armidale, NSW, Australia
[2] School of Veterinary Science, University of Melbourne, Melbourne, VIC, Australia

## ABSTRACT

The social context of social species such as sheep can modify their physiological and behavioural responses to stressors, through social buffering and social facilitation. Social buffering can lead to amelioration of stress, while social facilitation can lead to stress responses in an observer animal in the presence of a conspecific in distress. The current study investigated twin lambs undergoing ring castration, grouped either homogeneously with a castrated lamb (actor), or heterogeneously with a non-castrated lamb (observer) to examine the impact of social grouping on behavioural responses. Each lamb was scored for two classes of behaviour: pain-related behaviours and postures that are putatively related to the response to castration; and attentional behaviours directed at its twin. Thus, each lamb participated in the experiment as an actor exhibiting pain-related behaviours and postures, and as an observer of its twin. When behaviours of lambs were assessed as actors, there was a significant ($P < 0.05$) effect of castration but no significant effect of social grouping on expression of pain-related behaviours. When behaviours of lambs were assessed as observers, homogeneous grouping of castrated lambs increased attention towards the other twin in comparison to non-castrated lambs grouped homogeneously or lambs grouped heterogeneously ($P < 0.01$). Non-castrated lambs grouped homogeneously demonstrated significantly lower numbers of head direction changes ($P < 0.001$) and lower number of ear posture changes ($P < 0.05$) than heterogeneously grouped or castrated lambs. This study indicates that social attention between twin lambs is not clearly dependent on pain status of the actor lamb. It suggests that in order for the observer lamb to provide significant attention to the actor lamb displaying pain-related behaviour, the observer lamb also needs to be experiencing pain concurrently. Furthermore, there is some evidence that the presence of pain-related behaviours can lead to increased attention to the surrounding environment in non-castrated lambs. Understanding the effect of concurrent experience and varying social context assists us to improve our understanding of results of other experiments on pain-related behavioural responses.

Corresponding author
Alison Small, alison.small@csiro.au

## INTRODUCTION

Understanding social influences on behaviour is a vital component to measuring welfare states in social animals such as sheep (*Colditz, Paull & Lee, 2012*). It is currently understood that pain changes the animal's physiology and behaviour (*Molony & Kent, 1997*) and the behaviour of animals can be influenced by their social context (*Dall et al., 2005*). Thus, alteration of the animal's behaviour in response to pain can also be dependent on the social context. Animals experiencing pain may invoke similar pain-related behavioural responses in bystander animals that have not received physical trauma via a process such as social facilitation (also called social contagion) (*Nicol, 1995*). Social facilitation is a term used to define a situation where the behaviour of one individual induces the same behaviour in another individual (*Nicol, 1995*).

Alternatively, the intensity of the behavioural and physiological responses of an animal to a physical trauma such as castration could be reduced by the presence of non-traumatised pen-mates by a mechanism such as social buffering (*Nicol, 1995*; *Hennessy, Kaiser & Sachser, 2009*; *Kikusui, Winslow & Mori, 2006*; *Veissier et al., 1998*). Since it is advantageous for the individuals to engage in behaviour to promote survivability of others with similar gene composition (*Hamilton, 1964*), it is no surprise that ewes are more attentive to their offspring experiencing pain (*Hild et al., 2011*). The mechanism of increased attention may be due to emotional contagion, where a convergence of emotional states between individuals occurs and this results in increased maternal care (*Hild et al., 2011*). Social buffering of pain behaviours in lambs has been demonstrated to depend on the relationship between the actor and observer (*Guesgen et al., 2014*). For this reason, our study utilised twin lambs. Social buffering of the physiological and behavioural response to stress has also been demonstrated by providing a picture of a conspecific to isolated sheep (*Da Costa et al., 2004*). Social support plays an important role in buffering the effects of stress and has important implications for farm animal welfare as reviewed by *Rault (2012)*.

Behavioural indicators of pain in lambs in response to painful husbandry procedures are well established. These include changes in active pain-related and postural behaviours (*Paull et al., 2012*), the number of ear posture changes (*Guesgen et al., 2016*) and attention (*Hild et al., 2011*).

This study investigated whether the social context of twin siblings influences the behavioural responses to pain. We predicted that in twins, the presence of a non-castrated lamb (observer lamb) with a castrated lamb (actor lamb) would reduce the expression of pain-related behaviours in the actor lamb through social buffering, while an observer lamb would demonstrate increased pain-related behaviours in the presence of a castrated sibling through social facilitation.

## MATERIALS AND METHODS

The experiment was performed at CSIRO's FD McMaster Laboratory, Armidale, NSW, Australia and was approved by the Institutional Animal Ethics Committee, ARA 14/14.

The study followed methodology previously utilised in a similar study (*Colditz, Paull & Lee, 2012*), as outlined below.

## Animals and treatments

Forty-eight male fine wool Merino lambs born as twin lambs and their mothers were used for this experiment. Lambs were born in spring of 2014 at pasture following natural mating. Dates of birth were not recorded at birth, so mother-offspring and twin pairs were identified in the field by observation of proximity, following and suckling behaviour at the end of the 5-week lambing period. Selection criteria were that the ewes and lambs appeared clinically healthy, lambs were vigorous in following the ewe and the lamb suckling was well tolerated by the ewe. Animals were inducted into the animal house in the week prior to the trial to acclimatise to the treatment environment, and they were fed sheep pellets (Ridley Agriproducts, Tamworth, Australia; 20.6% crude protein DM; 12.5 MJ/kg DM) at a rate of 0.8 kg/dry sheep equivalent/day and 100 g/day oaten chaff. Lambs had an age between 6 and 12 weeks and bodyweights ranging from 14.0 to 22.8 kg on the day prior to treatment. In addition, lambs were weighed on entry to the animal house and divided into 3 cohorts of 16 lambs, stratified by average weight of the twin pair. Within each cohort, twin pairs were randomly allocated to treatment groups. During acclimation to the animal house environment, each cohort of 16 lambs were housed in a large group pen together with their mothers. On the day prior to treatment the lambs underwent a clinical examination and blood sampling for health screening, were weighed, the identification marks applied to the flanks, and placed into their assigned treatment groups of two ewe-lamb-lamb units into each observation pen. Mother-offspring units in which any animals did not appear clinically healthy or did not return normal values of heart rate, respiratory rate, rectal temperature and haematology including differential white blood cell count on the day prior to treatment were to be excluded from the study: this did not occur.

Four treatment pens were used for each cohort, each treatment pen consisting of two ewes with their respective twin offspring. Both twin pairs in each treatment pen received the same treatment as follows: homogeneous grouping (both twins ring castrated (CC, positive control) or both twins non-castrated (NN, negative control)), heterogeneous grouping (ring castrated lamb observing a non-castrated lamb (CN) and non-castrated lamb observing a castrated lamb (NC)), such that each cohort contained two pens of heterogeneous pairings, and one each of non-castrated and castrate homogeneous pairings. That is, there were 4 NC pairs per cohort, so a total of 12 NC pairs in the analysis. Within each pairing, both lambs were evaluated as both 'observer' and 'actor' for the purposes of the study (Fig. 1), leading to $n = 12$ in each observer-actor unit (C observing C, N observing N, N observing C, C observing N). Ewe-offspring groups were randomised to treatment group by blinded selection of a coloured marble from a cloth bag, the marble colour being pre-assigned to treatment using a similar approach.

Cohorts were tested sequentially over 9 days, such that on the completion of one cohort of observations, those animals were removed from the treatment pen on the subsequent morning and the pen cleaned (Table 1). On the following day, the next cohort was prepared and placed into treatment pens, so that they adjusted to the treatment pens overnight with treatment applied the following day.

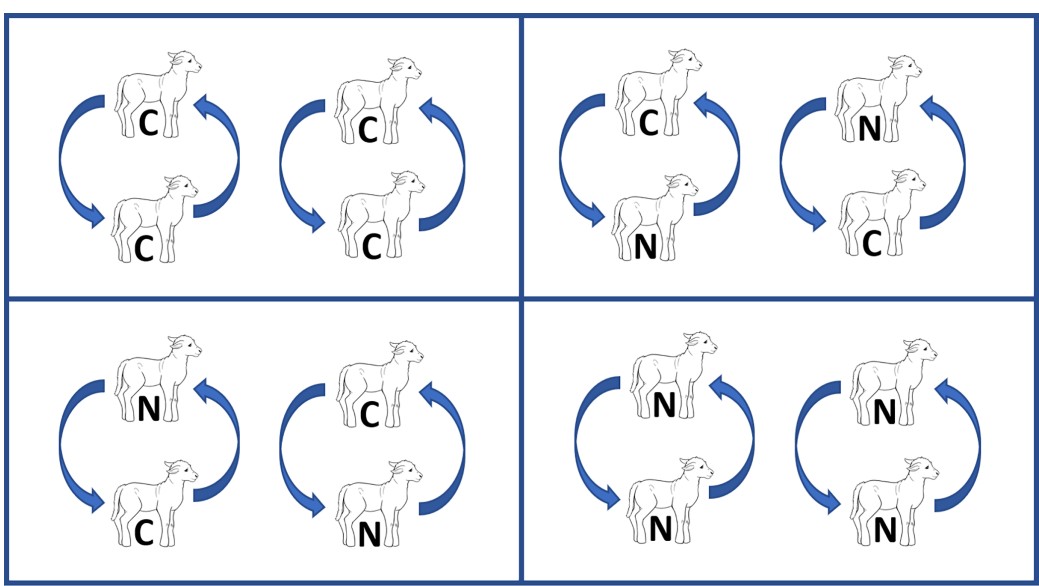

**N** not castrated; **C** castrated;
arrow shows direction of observation from observer lamb to actor lamb

**Figure 1 Schematic diagram of study design.** Diagram of lamb groupings in each of four pens in each cohort. C: castrated lamb; N: non-castrated lamb. Curved arrow shows direction of attention from observer lamb to actor lamb.

Allocation of treatment group pairings to treatment pens were different for each cohort to control for potential pen effects. The pens were as described by *Colditz, Paull & Lee (2012)*: 4.4 × 3.0 m with slatted wooden floor, with visual contact between pens limited through use of opaque hessian cloth and rubber mats on the vertical surfaces, but animals could still hear one another between the social groups. The order of treatment of pens was selected to minimise distraction to groups that had previously been treated, and behaviour observations were underway.

Lambs were placed in dorsal recumbency in a lamb marking cradle for approximately 1 min for their treatment. Ring castration was performed according to the standard industry practices (*Lloyd & Playford, 2013*), without analgesia or anaesthetic drugs. Non-castrated lambs had their scrotum manipulated for a similar period of time to the time required to apply the elastrator ring, without a ring being applied. Within the heterogeneous pairings, the selection of the castrate vs. the non-castrate animal in each pair was randomised by blind selection of a coloured marble from a cloth bag. Treatments were administered approximately between 08:45 and 09:30 h with each pair of twins treated every 5 min to allow time for ear posture video recording of each ewe-offspring group.

Continuous video footage of each pen was captured from 08:00 to 14:00 h by two video cameras per pen (Model SCC-B2313P; Samsung Day And Night Digital Color Camera, no city disclosed by manufacturer, China). Additionally, 1 min each of twin lambs and the respective ewe was captured by hand-held digital video recorder (Model HDR-XR260E; Sony Handycam, no city disclosed by manufacturer, Japan) 10 min after the lamb was

| Time period | Activity |
|---|---|
| **Table 1** | **Outline of study timeline.** |
| Week-2 | Identification of ewe-offspring groupings by observation of proximity, following and suckling behaviour in the field, mustering of ewe-offspring groupings, supplementation of grass feed with animal house pellets.<br>Animals checked daily. |
| Week-1 | Ewe-offspring groupings moved into animal house, lambs weighed, ear tags recorded, assigned to cohort based on average bodyweight in each twin pair.<br>Animals fed and checked daily. |
| Day-1 | Cohort 1 lambs weighed, clinical examinations performed and blood samples assayed, ewe-offspring groups assigned to treatment pen, identification marks applied to flanks of lambs. |
| Day 0 | Animals fed at 08:00, remote video recording started<br>08:45–09:30 treatments applied; 1 min of hand-held video footage of each animal in each ewe-offspring group collected starting 10 min after the lamb was returned to the pen;<br>14:00—remote video recording stopped and downloaded, animals checked. |
| Day 1 | Cohort 1 removed from treatment pens, treatment pens cleaned. |
| Day 2 | Cohort 2 lambs weighed, clinical examinations performed and blood samples assayed, ewe-offspring groups assigned to treatment pen, identification marks applied to flanks of lambs. |
| Day 3 | Animals fed at 08:00, remote video recording started<br>08:45–09:30 treatments applied; 1 min of hand-held video footage of each animal in each ewe-offspring group collected starting 10 min after the lamb was returned to the pen;<br>14:00—remote video recording stopped and downloaded, animals checked. |
| Day 4 | Cohort 2 removed from treatment pens, treatment pens cleaned. |
| Day 5 | Cohort 3 lambs weighed, clinical examinations performed and blood samples assayed, ewe-offspring groups assigned to treatment pen, identification marks applied to flanks of lambs. |
| Day 6 | Animals fed at 08:00, remote video recording started<br>08:45–09:30 treatments applied; 1 min of hand-held video footage of each animal in each ewe-offspring group collected starting 10 min after the lamb was returned to the pen;<br>14:00—remote video recording stopped and downloaded, animals checked. |
| Day 7 | Cohort 3 removed from treatment pens, treatment pens cleaned.<br>All animals returned to farm. |

placed back in the pen for ear posture assessment. From this footage, 30 s of video was selected for ear posture analysis per animal. However, due to the poor quality of video obtained from the handheld position of the camera, it was not possible to identify specific ear postures as described by others (*Boissy et al., 2011*; *Reefmann et al., 2009a*), therefore only data on the number of times the ear posture changed was collected. Data collection was carried out by a single observer, blinded to treatment, and a subset of two videos from each cohort were assessed by a second blinded observer to verify reliability of counting ear posture changes. Animal were fed at 08:00 h, the feeding process taking 10 min, so that observations were not modified by feeding behaviours. In the following day, the ewes and lambs were removed from the study pens, the pens were cleaned and the next cohort of sheep were moved into the pens.

## Behavioural data and statistical analyses

Each lamb within a pen was identified by one of four symbols marked on the wool of the lamb using spray marker (Steadfast Stockmark, Dymark, Australia). Allocation of symbols

**Table 2 Behaviour as actors: description of postural behaviours and active pain-related behaviours recorded during the experiment, as described by Colditz, Paull & Lee (2012).**

| Behaviour | Abbreviation | Description |
|---|---|---|
| *Postural behaviours* | | |
| Normal upright | Nu | Standing, walking or playing while exhibiting a usual posture or gait; smooth movements. |
| Abnormal upright | Au | Standing exhibiting unusual posture for example Rounded, hunched appearance; ataxia; jerky movements; walking unsteadily, backwards, on knees. |
| Total upright | Sum of: Nu+au | All upright postures pooled. |
| Normal lying | Nl | Ventral recumbency, all legs tucked under body or very close to body. |
| Abnormal lying | Al | Twisted lying; ventral recumbency with forelimbs tucked under body, one or both hind limbs partially or fully extended; including dog sitting and lateral lying (lateral recumbency with one shoulder on ground, hind limbs and/or forelimbs fully extended). |
| *Active pain-related behaviours* | | |
| Restlessness | Rst | Number of times lamb stood up and laid down. Instances of lamb rising as far as its knees included in the one count. |
| Kicking/foot stamping | Fsk | Either a front or hind limb (usually hind limb) was lifted and forcefully placed on the ground while standing or was used to kick while standing or lying. |
| Rolling | Rl | Rolled from lying on one side to the other without getting up. Half rolls where the lamb rolled on its back and then returned to lying on the same side included. |
| Jumping | Jmp | All four feet off ground simultaneously. |
| Licking/biting wound site | Lbw | Movement of the head beyond the shoulder, including both looking and touching at the source of pain and grooming. |
| Head shake | Sh | Forceful voluntary shake of the head. |
| Easing quarters | Eq | Abnormally lowers rear quarters (standing) or attempts to keep quarters off the ground (lying). |
| Sum of pain-related behaviours | Sum of: Rst+fsk+rl +jmp+lbw+eq | All pain related behaviours pooled. |

were randomised so that the observer was blinded to castration treatment. Video footage of each lamb was recorded for 5 h after treatment applications were completed, using a digital video recorder (Pacific Communications, Port Melbourne, VIC, Australia), but for the current study, attention was focused on the first 45 min post castration. The ethogram used in this study was based on behaviour patterns described in previous studies (Grant, 2004; Kent, Molony & Graham, 1998; Colditz, Paull & Lee, 2012) of behavioural responses of lambs following ring castration. Detail of the ethogram, the behaviour codings used and the calculations used to determine combined totals (e.g. total upright and Sum of Pain-Related Behaviours) is presented in Table 2. Video footage was assessed offline using SmartViewer software (Hanwha Techwin America, Teaneck, NJ, USA). Counts of postural behaviours and active pain-related behaviours performed by each lamb in a continuous block of 30 min (15–45 min after treatment) were manually collected in MS Excel (Microsoft Inc., Redmond, WA, USA) by a single observer. A subset of two videos from each cohort were assessed by a second blinded observer to verify reliability of data collection. The attention direction behaviours are described as shown in Table 3. Attention direction behaviours were categorised as the time spent looking towards specific

**Table 3 Behaviour as observers: description of attention direction behaviours recorded during the experiment.**

| Attention behaviour | Abbreviation | Description |
|---|---|---|
| Looking at ewe | Ewe | Head and eyes turned in the direction of the ewe |
| Looking at twin | Twin | Head and eyes turned in the direction of its twin sibling |
| Looking at own tail | Tail | Head and eyes turned in the direction of its own tail |
| Looking elsewhere | Other | Head and eyes turned elsewhere than one of the above |
| Head direction changes | Hchanges | The number of times head position changed from one of the above to another |

targets as described by *Monk et al. (2019)* and were observed for seven 1-min periods at 5 min intervals from 15 to 45 min post castration.

Results for postural behaviours, active pain-related behaviours and ear posture changes were calculated as counts of each behaviour, and results for attention behaviours were calculated as sum of time spent looking towards specific targets.

Data were collated in MS Excel (Microsoft Inc., Redmond, WA, USA) analysed using ASREML (VSN International, Hemel Hempstead, UK), using a linear fixed-effects regression model including treatment, cohort, pen and first order interactions. Where necessary, behavioural data were normalised for analysis of variance. Logarithmic transformation was required for the attention direction 'Looking at own tail' and all postural and pain-related behaviours. In some categories of behaviour, there were insufficient data for analysis therefore these were pooled into the 'Sum of Pain-Related Behaviours' category. Rolling (Rl), jumping (Jmp) and headshake (Sh) were not analysed individually due to low frequencies but were used to calculate total abnormal behaviours. 95% confidence intervals were calculated as the mean ± 1.96 SEM to determine treatment differences; $P < 0.05$ was considered significant.

# RESULTS

## Behaviour as actors: pain-related and postural behaviours

There was a significant ($P < 0.05$) effect of castration treatment on Sum of Pain-Related Behaviours (Table 4), and a significant effect of cohort on normal lying (Nl), kicking/foot stamping (Fsk); and Licking/biting wound site (Lbw). However, there was no significant effect of social grouping on any pain-related behaviours and no significant interactions between treatment and pen, treatment and cohort or pen and cohort.

## Behaviour as observers: attention direction

There was a significant ($P < 0.05$) effect of treatment on attention to twin, attention to tail, attention elsewhere and changes in attention direction, and no significant interactions between treatment and pen, treatment and cohort or pen and cohort. For looking at ewe, there were no significant main effects or interactions (Table 5). When both twin siblings were castrated, they displayed increased time looking at the other twin

**Table 4 Behaviour as actors (n = 48): effect of grouping and treatment on pain-related and postural behaviours of lambs following treatment.** Counts of postural behaviours and pain-related behaviour was recorded continuously from 15 to 45 min post-treatment. All data presented are log transformed least squares means with back transformed data in parentheses. There were no significant first-order interactions between pen, cohort and treatment. [a,b]Means across each row with different superscripts are significantly different ($P < 0.005$).

| Grouping | Homogeneous | | | | | Heterogeneous | | | | |
|---|---|---|---|---|---|---|---|---|---|---|
| Actor lamb status | Castrated | | Not castrated | | | Castrated | | Not castrated | | |
| | Mean (back transformed estimate) | SE | Mean (back transformed estimate) | SE | P | Mean (back transformed estimate) | SE | Mean (back transformed estimate) | SE | P |
| Postural behaviours | | | | | | | | | | |
| Normal upright | 1.34 (21.05)[a] | 0.05 | 0.33 (1.12)[b] | 0.05 | <0.001 | 1.35 (21.51)[a] | 0.05 | 0.33 (1.14)[b] | 0.05 | <0.001 |
| Abnormal upright | 0.57 (2.70)[a] | 0.06 | 0 (0)[b] | 0.06 | <0.001 | 0.54 (2.44)[a] | 0.06 | 0 (0)[b] | 0.06 | <0.001 |
| Total upright | 1.40 (23.87)[a] | 0.05 | 0.33 (1.12)[b] | 0.05 | <0.001 | 1.41 (24.51)[a] | 0.05 | 0.33 (1.14)[b] | 0.05 | <0.001 |
| Normal lying | 1.25 (16.83)[a] | 0.06 | 0.19 (0.55)[b] | 0.06 | <0.001 | 1.23 (16.06)[a] | 0.06 | 0.09 (0.23)[b] | 0.06 | <0.001 |
| Abnormal lying | 1.07 (10.63)[a] | 0.05 | 0 (0)[b] | 0.05 | <0.001 | 0.99 (8.82)[a] | 0.05 | 0 (0)[b] | 0.05 | <0.001 |
| Active pain-related behaviours | | | | | | | | | | |
| Restlessness | 1.43 (26.08)[a] | 0.05 | 0.04 (0.10)[b] | 0.05 | <0.001 | 1.38 (22.99)[a] | 0.05 | 0.08 (0.19)[b] | 0.05 | <0.001 |
| Kicking/foot stamping | 0.79 (5.17)[a] | 0.11 | 0.08 (0.19)[b] | 0.11 | <0.001 | 0.81 (5.44)[a] | 0.11 | 0.03 (0.06)[b] | 0.11 | <0.001 |
| Licking/biting wound site | 0.78 (5.00)[a] | 0.09 | 0.23 (0.72)[b] | 0.09 | <0.001 | 0.56 (2.62)[a] | 0.09 | 0.13 (0.33)[b] | 0.09 | <0.001 |
| Easing quarters | 0.35 (1.21)[a] | 0.06 | 0.09 (0.23)[b] | 0.06 | =0.005 | 0.35 (1.21)[a] | 0.06 | 0.08 (0.19)[b] | 0.06 | =0.005 |
| Total pain behaviour | 1.63 (41.60)[a] | 0.07 | 0.41 (1.60)[b] | 0.07 | <0.001 | 1.56 (35.53)[a] | 0.07 | 0.26 (0.80)[b] | 0.07 | <0.001 |

**Table 5 Behaviour as observers (n = 48): effect of treatment on attention directed behaviour of lambs for 45 min following treatment.** Data are mean counts and collected continuously for 1 min at 5 min intervals between 15 and 30 min post-treatment. [a,b,c]Means across each row with different superscripts are significantly different ($P < 0.01$). NS: not significant. There were no significant first order interactions between pen, cohort and treatment. [#]Data presented are log transformed least squares means with back transformed data in parentheses.

| Grouping | Homogeneous | | | | | Heterogeneous | | | | |
|---|---|---|---|---|---|---|---|---|---|---|
| Actor lamb status | Castrated | | Not castrated | | | Castrated | | Not castrated | | |
| | Mean (back transformed estimate) | SE | Mean (back transformed estimate) | SE | P | Mean (back transformed estimate) | SE | Mean (back transformed estimate) | SE | P |
| Attention direction | | | | | | | | | | |
| Looking at ewe | 64.25[a,b] | 8.92 | 55.67[a,b] | 8.92 | NS | 77.50[a,b] | 8.92 | 50.25[b] | 8.92 | NS |
| Looking at twin | 42.50[a] | 4.35 | 18.00[b] | 4.35 | =0.006 | 23.42[b] | 4.35 | 27.75[b] | 4.35 | NS |
| Looking at own tail[#] | 0.74 (5.53)[a] | 0.11 | 0.08 (1.76)[c] | 0.11 | =0.002 | 0.50 (3.16)[a,b] | 0.11 | 0.25 (1.20)[b,c] | 0.11 | NS |
| Looking elsewhere | 245.25[a] | 9.32 | 291.42[b] | 9.32 | =0.011 | 255.58[a,b] | 9.32 | 275.08[a,b] | 9.32 | NS |
| Head direction changes | 34.08[a] | 2.40 | 17.67[b] | 2.40 | <0.001 | 35.08[a] | 2.40 | 27.75[a] | 2.40 | NS |

($F_{3,33} = 5.04$, $P = 0.006$). When twins were homogeneously grouped, the castrated lambs showed increased time looking at their own tails than non-castrated lambs ($F_{3,33} = 6.17$, $P = 0.002$). When both twin siblings were non-castrated, they demonstrated

significantly lower number of head direction changes as compared to heterogeneously grouped twins, or those where both twins were castrated ($F_{3,33} = 10.27$, $P < 0.001$).

### Ear posture changes

Other than a decreased number of ear posture changes ($F_{3,33} = 3.67$, $P = 0.022$) displayed when both twin siblings were non-castrated, there were no differences between treatments, no significant main effects and no significant interactions between treatment and pen, treatment and cohort or pen and cohort.

## DISCUSSION

In this study, lambs undergoing castration were grouped either homogeneously with a castrated twin, or heterogeneously with a non-castrated twin to examine the impact of social grouping on behavioural responses to painful husbandry practices. Regardless of social grouping, castrated lambs showed a significant increase in all pain-related behaviours as well as increased changes in posture. These differences are consistent with previous studies that have investigated behavioural responses of lambs to painful husbandry procedures (*Grant, 2004*; *Colditz, Paull & Lee, 2012*; *Paull et al., 2012*).

While it was predicted that the presence of a pain-free twin would provide social buffering to reduce pain-related behaviours in castrated lambs, this was not found to be the case. The results indicate that the presence of a twin sibling with a different treatment did not substantially modify the responses to the castration treatments via social buffering or contagion. Similarly, *Colditz, Paull & Lee (2012)* found that heterogeneous mixing of lambs that were sham handled, surgically castrated or ring castrated did not substantially alter the behavioural responses to the specific castration treatment. There may, however, be an effect of the specific relationship between the lambs that was not possible to examine in our study as all subjects had both a twin and ewe present. A recent study comparing social buffering of pain behaviours in response to tail docking showed significant differences between twin lambs and unrelated lambs where the pairs of twins displayed lower frequency of kicking and rolling than pairs of unrelated lambs (*Guesgen et al., 2014*). As the current study's results show no behavioural difference between homogeneous grouping (CC) and heterogeneous grouping (CN or NC), it can be suggested that the effect of social buffering is more dependent on the presence of a relative rather than the pain status of the observer lamb. Interactions between treatment (castrated or non-castrated) and social grouping were not significant for any pain-related and postural variables.

It cannot be ruled out that the sensitivity of the behavioural measures used in our study may not have been high enough to detect subtle behaviours displayed by twin lambs. Other potential measures such as distance between lambs and facial expressions should be investigated in future studies. Furthermore, we cannot exclude the dam's influence on pain behaviour (*Hild, Andersen & Zanella, 2010*) which may have masked the social interaction between lambs.

In addition to the analysis of active pain-related behaviour and posture of lambs as actors, attention directed behaviour and ear posture changes of lambs as observers were

investigated as additional indicators between twins of different social grouping. When both twins were castrated, twins showed a significant increase in the amount of time spent looking at their twin in comparison to all other social groupings. This indicates that the observer lamb was more attentive to its twin sibling when in pain, only when the observer lamb itself was also simultaneously experiencing pain. Attentiveness has been previously measured in ewes towards lambs that are in pain, and exploring and glancing have been identified to be directly correlated with the level of lamb's pain behaviours (*Hild et al., 2011*). This is consistent with our finding that attention direction behaviour is increased when both twin lambs are castrated and in pain but there were no differences in attention direction behaviour when only one twin was in pain. It may be that for lambs to show increased attention towards a twin in pain, they need to be experiencing the state of pain concurrently. This may be a form of emotional empathy as the lamb is reacting to the observed experiences of the other. Related chickens have been demonstrated to be capable of emotional empathy with mother hens being responsive to their chicks' distress (*Edgar et al., 2011*), but emotional empathy was not found in unrelated hens (*Edgar et al., 2012*).
It may also be that emotional contagion is occurring with a convergence of emotional states between the lambs as has been suggested as a mechanism for increased attention by ewes towards castrated lambs (*Hild et al., 2011*).

The finding of decreased head direction changes (changes in attention) when both twins were non castrated aligns with the ear posture changes which were significantly lower in non-castrated twin siblings in comparison to other social groups. This suggests that non-castrated lambs placed with other non-castrated lambs in the same pen environment have lower alertness or attention towards their surroundings in comparison to castrated lambs. Conversely, this highlights that a non-castrated lamb placed with a castrated lamb in pain demonstrated heightened alertness to the surrounding environment, similar to the castrated sibling.

It is known that acute stress and pain induces activation of the sympathetic nervous system and hypothalamic-pituitary-adrenal (HPA) axis (*Tsigos & Chrousos, 2002*), resulting in release of corticotropin-releasing hormone (CRH). CRH in the hypothalamus appears to mediate anxiogenic behaviour including heightened alertness (*Reefmann et al., 2009b*; *Reefmann, Wechsler & Gygax, 2009*). Indeed, pharmacological induction of anxiety altered attention and vigilance behaviours in sheep (*Monk et al., 2018*) and hence, increased head direction and ear posture changes in all castrates was an expected result. However, the mechanism of social influences on pain and stress from castrated lamb to non-traumatised lamb leading to increased alertness is unclear.

It is possible to attribute the influence of behaviour of heightened attention to an instinctive evolutionary mechanism to increase the probability of survival in nature (*Decety et al., 2016*). As mentioned in the introduction, it can be seen as a behavioural mechanism to increase the preservation of genes by increasing the chance of survival of individuals with similar genetic make-up. However, due to the limitations of this experiment, it is difficult to conclude whether this social influence on attention directed behaviour in response to pain is specific to twin lambs.

This was the first study to utilise handheld video recording to investigate ear postures in a pen situation. While distinct ear postures have been linked with differing emotional states in sheep (*Boissy et al., 2011*; *Reefmann et al., 2009a*), unfortunately, it was very difficult to capture footage of both ears at the same time, and the video was not recorded from enough height to perceive the depth of the ears' position in relation to the skull. However, it was possible to identify ear posture changes, hence only the data for ear posture change was presented. A study in lambs undergoing the painful procedure of tail docking reported that an increased number of changes in ear postures was observed in response to tail docking (*Guesgen et al., 2016*), however we observed few differences in ear posture changes in the current study. Use of the hand-held recording technique may have affected the behaviour of the animals, although the cameraperson made an effort to move slowly around the edges of the pen and remain unobtrusive. Prior to treatment, the animals had been subject to 1 week of acclimation to the animal house environment including movement of personnel in and out of the pens to reduce the effect of personnel activity on the animals' behaviour.

Further research is required to explore the effects of animal temperament, breed type and age of lamb on social transmission of behaviours. In terms of age, the lambs in this study ranged from 6 to 12 weeks of age, after the reported period in which rapid change in behavioural responses to pain occurs (*Guesgen et al., 2011*; *Johnson et al., 2009*; *McCracken et al., 2010*).

## CONCLUSIONS

This study found no evidence of social buffering in twin lambs with their mother present as there was no significant decrease in pain-related behaviours and postures in lambs in the company of a pain free twin. There is some suggestion that in order for the observer lamb to provide significant attention to the actor lamb displaying pain-related behaviour, the observer lamb also needs to be experiencing pain concurrently. Furthermore, there is some evidence that social transmission of pain behaviours may lead to increased attention towards the surrounding environment. Understanding the effect of concurrent experience and varying social context assists us to improve our understanding of results of other experiments on pain-related behaviour.

## ACKNOWLEDGEMENTS

We thank Dominic Niemeyer, Troy Kalinowski, Tim Dyall, Aymeric de Trogoff du Boisguezennec and Anna Wilson for technical assistance.

### Funding

This work was supported by Australian taxpayers through federal government appropriation funding of CSIRO. Andrew Cho was supported by the CSIRO summer student scholarship programme 2014 (Andrew Cho). The funders had no role in study design, data collection and analysis, decision to publish, or preparation of the manuscript.

## Grant Disclosures

The following grant information was disclosed by the authors:
CSIRO.

## Competing Interests

The authors declare that they have no competing interests.

## Author Contributions

- Andrew Inhyuk Cho performed the experiments, analysed the data, prepared figures and/or tables, authored or reviewed drafts of the paper, and approved the final draft.
- Caroline Lee conceived and designed the experiments, analysed the data, authored or reviewed drafts of the paper, and approved the final draft.
- Alison Small conceived and designed the experiments, performed the experiments, analysed the data, authored or reviewed drafts of the paper, and approved the final draft.

## Animal Ethics

The following information was supplied relating to ethical approvals (i.e. approving body and any reference numbers):

The experiment was approved by the CSIRO's FD McMaster Laboratory, Armidale, NSW, Australia Institutional Animal Ethics Committee, ARA 14/14.

## Data Availability

The raw data are available in the Supplemental File.

## Supplemental Information

Supplemental information for this article can be found online at http://dx.doi.org/10.7717/peerj.10081#supplemental-information.

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
