# Peer review of "Attention behaviours but not pain-related behaviours are modified by the presence of a twin in lambs undergoing castration by rubber ring"

_PeerJ, doi:10.7717/peerj.10081_

## Round 0.1 · original submission · Major Revisions

· Academic Editor

Major Revisions

The manuscript in its current state needs major revisions with attention to consistency in terminology and data and results presented, more details in your methods, adjusting for confounders (example age, if you believe there wasn't then what is the evidence for lack of confounding by age and other confounders), what measures were taken to reduce bias in observations and data measurements and justify the study design in terms of measuring behaviors using in person phone recordings (what measures were taken to account for any bias this may introduce).

Reviewer 1 ·

Basic reporting

The article is concisely written, and the format and data submitted meet the standards of PeerJ. There are a few areas where clarity of the reporting could be improved.

1. Were both members of all pairs observed? The description of treatments (lines 82-83) makes it sound as if only one individual from the heterogeneous pairs was considered as an ‘observer’ and one as an ‘actor’, but Tables 3 and 4 indicate that all 48 lambs were included in both.
2. Some of the terminology used may not be the most appropriate. The term “social facilitation” is used in the abstract only, and the term “social transmission” elsewhere; both of these, in my experience, are used to describe an increase in performance of a behaviour due to presence of or performance by conspecifics. Since you are also looking at decreases in performance of certain behaviours through buffering, this is a little confusing. The label “pain avoidance” behaviours is also unusual – the behaviours listed are common indicators of pain, but do not necessarily help the animal avoid pain. The term “pain related behaviours” used in the title is preferable.
3. A little more discussion to place this in context would be helpful. For example, what does this add to what was known from the Guesgen et al. study mentioned in the discussion? Is there other evidence of these phenomena in grazing animals? More information/elaboration on the ideas raised in the introduction would help clarify your aims and use of these terms. Rault’s review of social support may be useful.
4. The stated aim also does not really cover effects in both directions. A clear statement of predictions is desirable.

Experimental design

I have a few concerns regarding the experimental design, as follows:
1. The rationale for the choice of treatments is somewhat unclear. It appears that you could test whether social buffering OR contagion occurs, but it would be difficult to say which because there is no true control without any social effect. The degree to which these effects can be tested depends on how the partners in heterogeneous pairs were used (see question above); no explanation is given for what you intended to test by including the non-castrated actors. Was social contagion expected in non-castrated animals?
2. If only 3 sets of each type of heterogeneous pair were used (1/cohort), is this enough to draw inferences about those situations?
3. Is there evidence to support the use of number of ear movements for this purpose? And is 1 min of observation sufficient?
4. Line 97 – Was this always the same time post-procedure?

Appropriate consideration was given to randomization and pen effects, and it is good to see use of blinding.

Validity of the findings

1. The stated aims were to see if social effects decrease distress and reduce predation risk (line 59). However, these are not directly returned to in the discussion. The use of these measures to make inferences about distress was not fully justified, and any link to predation should be explained for these specific results.
2. The analysis does not appear to control for age (unless this is the meaning of “day”?) although there is a large range. Could any environmental factors such as weather have influenced results?

Conclusions were well stated.

Additional comments

1. What was the aim of using only twins, given that most received the same treatment? The Discussion suggests relatedness may be important but it is not clear if this was intentional, or whether it differs from other work.
2. Line 68 is missing punctuation.
3. Line 94 was duration of this manipulation similar?
4. Was reliability of counting ear posture changes checked, given the difficulty in observing postures?

·

Basic reporting

This work is original and within the aims and scope of the journal. The paper studied the effect of grouping castrated lambs either with castrated or non-castrated twins to examine the effect of social grouping on modifying the pain-related behaviors. This research adds more information about the theory of social and emotional contagion. However, observing the behavior of animals alone is not enough to answer the research questions. The paper is not well written and missing important information to judge the obtained results. For example, as it stands the materials and methods are very difficult to read and it is not clear exactly what the authors are trying to research. The conclusion needs to be more succinct and focused on the purpose of this study. The authors should be careful in their conclusion since they had insufficient data for analysis and low frequencies of some behavioral patterns. There is no discussion of the limitations of the study, although there are many, particularly with regard to the recording of behaviors and missing data. Also, no results for interactions are reported in tables.

Experimental design

The way that materials and methods are written is very confusing and hard to digest. I suggest the authors rewrite it and provide a diagram or graphical abstract describing the experimental design. It is not clear how the research will fill the knowledge gap. Please provide a clear statement regarding the identified research gap.

Here are more detailed comments.

Introduction:
L44: Please provide a reference for your statement
L49: Please provide more details about social contagion for readers who are not familiar with this theory. I suggest that you provide more detail regarding emotional contagion such as Hild et al. (2011), and social buffering (Da Costa et al. 2004)
L58: It is not clear how the research will fill the knowledge gap. Please provide a clear statement regarding the identified research gap
Materials & Methods:
L72: Please provide more detailed information about animals such as age, BW at the beginning of the experiment. What are the criteria used to identify animals for this experiment? Please clarify?
L76: You had lambs with different ages. How you adjusted for age difference in your treatment groups?
L79: What is the difference between pens and treatment pens. This is very confusing and hard to digest. Previously you mentioned that 16 lambs were housed in a pen and here you are saying 2 ewes with their twin. I suggest the authors provide a diagram or graphical abstract describing your experimental design
L83: How authors identified which lamb observing which? How did you decide which lamb will be castrated and which are not?
L91: Change the 60s to 1 minute
L92: Which industry practices, please provide the reference
L96: What do you mean by “hand-held real time recording”?
L97: Which behaviors were recorded?
L102: How the ear posture was assessed and which references where used?
L106: Which type of markers used for the lambs?
L107: How many observers recorded the behavior of animals?
L108: How is the behavior observed for 5 hours after treatment? Which recording method was used? Please provide more detailed information about behavior recording and observation
L109: Pleases move the references following “previous studies”
L110: Which husbandry procedures? Please clarify
L114-115: Where are these criteria came from? Where are your references?
L105-116: Clear description of the recording of behavior data should be provided. For example, how is the output from the cameras was recorded and how many observers recorded the behavior of each animal. How you counted the posture of animals and avoidance behaviors? Is any software used for behavior recoding or it done manually? The behavior recording recorded in this study was not accurate and is insufficient for answering your research question.
Table 1: What is “No.” stand for, I guess it the number of. What is the purpose of providing abbreviations in the table for the ethogram? Please provide references in the table for measured behavioral patterns.
L125: Please specify what kind of model used, linear what? I suggest the author explore the effect of age in the model since the studied lambs had different ages.
L127: Specify what behavior data required normalization and how you did it?
L130: Here you mentioned that you calculated the 95%CI but there are no data reported in tables
L125-131: Please specify what significance level used to report treatment differences.

Validity of the findings

Overall, the data support the research question. Some questions need to clarify for example:

How as the data treated after obtained and what "sum of time for each behavior" means, is it total duration or bouts? Was the correlation between parameters examined? if so, how was it addressed? Did the author consider confounders such as animal personality in their models?

L141: This word doesn’t make sense “at the would site”. Did the author want to say “wound” instead of “would”?
L142: Where are the interaction data. Please add these data to tables
L146: “no significant interactions” with what?
L201: Behavior explicated by chickens may not exactly comparable to the behavior of sheep. I suggest that the author cite relevant references from the same species being studied.
L212: Provide reference, please
L226-230: This section should move to materials and methods. Regarding the handheld video recording, the general practice is to have cameras installed in the pen instead of being physically present which is interfere with the normal behavior of animals. I have a concern that the reported results could have been modified in the physical presence of observers.
Conclusions:
From the outcome of your research, it is not enough to make the current conclusion. Please rewrite.
Tables: Check the fonts for table headings. Add the results for different interactions tested in the model.

Additional comments

See above

---

## Round 0.2 · Minor Revisions

· Academic Editor

Minor Revisions

Thank you for addressing the reviewer comments, there are still a few minor revisions needed as identified by the reviewers that should get the text closer to publication if addressed.

Reviewer 1 ·

Basic reporting

The article now meets the journal's standards.
I would suggest clarifying in the introduction (as in the response to me) that the Guesgen et al. study actually showed pain effects on number of ear posture changes, not just the more familiar measure of different postures, to better justify the method here.

Experimental design

This does address an gap in research related to social context effects on buffering and/or contagion.

Validity of the findings

Conclusions regarding social buffering could perhaps be more clearly expressed, since the effects of presence of a twin in itself could not be tested here (because all subjects had a twin present, and ewe, which was the most welfare-friendly experimental design), and that is the typical social buffering paradigm. The MS as a whole I think should discuss more explicitly whether greater buffering effects should be expected if the companion is pain free themselves, which seems to have been one of the predictions. This was partially set up in the Introduction but never made explicit that I can see.

Other than that, conclusions are now appropriately limited and data and analysis seem sound.

Additional comments

Good work on the revisions. The MS is now much clearer and more logical.

Note that although you have changed "pain avoidance behaviours" through the text, you do not appear to have done so in the Table captions.

At line 249, an increase in "postural behaviours" does not make sense. I think what you mean is postural changes?

·

Basic reporting

The paper is too much improved, and authors have addressed all comments. The manuscript looks good to me and ready for publication. Just minor changes to be addressed:

1- Consider providing more information to figure 1. The figure should stand on its own. It will be nice to have pictures of lambs in this figure.
2- Don’t need an abbreviation column in Tables 2 and 3.
3- Consider making table 1 supplementary material.
4- Please consider presenting 95% CI in a separate column instead of having them in parenthesis, and use ± SE.

Experimental design

None

Validity of the findings

None

Additional comments

None

---

## Round 0.3 · accepted · Accept

· Academic Editor

Accept

Thank you for addressing the reviewers' and my comments, I am pleased to inform you that your manuscript has been accepted.
Best wishes,
Sharif